# COS-PPA: protocol to develop a core outcome set for primary progressive aphasia

Anna Volkmer ![ORCID],[1] David A Copland,[2,3] Maya L Henry,[4] Jason D Warren ![ORCID] ,[5] Rosemary Varley,[1] Sarah J Wallace ![ORCID] ,[2,3] Chris JD Hardy ![ORCID] [5]

[1]Department of Psychology and Language Sciences, University College London, London, UK
[2]Queensland Aphasia Research Centre, School of Health and Rehabilitation Sciences, The University of Queensland, Saint Lucia, Queensland, Australia
[3]Surgical Treatment and Rehabilitation Service (STARS) Education and Research Alliance, The University of Queensland and Metro North Health, St Lucia, Queensland, Australia
[4]Departments of Speech, Language, and Hearing Sciences and Neurology, The University of Texas at Austin, Austin, Texas, USA
[5]Dementia Research Centre, University College London, London, UK

**Correspondence to**
Dr Anna Volkmer;
a.volkmer.15@ucl.ac.uk

## ABSTRACT

**Introduction** The term primary progressive aphasia (PPA) describes a group of language-led dementias. Disease-modifying treatments that delay, slow or reverse progression of PPA are currently lacking, though a number of interventions to manage the symptoms of PPA have been developed in recent years. Unfortunately, studies exploring the effectiveness of these interventions have used a variety of different outcome measures, limiting comparability. There are more constructs, apart from word retrieval, that are important for people with PPA that have not received much attention in the research literature. Existing core outcome sets (COS) for dementia and non-progressive aphasia do not meet the needs of people with PPA, highlighting a need to develop a specific COS for PPA.

**Methods and analysis** This protocol describes a three-stage study to identify a COS for PPA interventions in research and clinical practice. The stage 1 systematic review will identify existing speech, language and communication measures used to examine the effectiveness of interventions for PPA in the research literature. Employing a nominal group technique, stage 2 will identify the most important outcomes for people with PPA and their families. The data collected in stages 1 and 2 will be jointly analysed with the project PPI group and will inform the stage 2 modified Delphi consensus study to identify a core outcome measurement set for PPA among a range of research disciplines undertaking intervention studies for people with PPA.

**Ethics and dissemination** Ethical approval for stage 2 of the study has been sought individually in each country at collaborating institutions and is stated in detail in the manuscript. Stage 3 has been granted ethical approval by the Chairs of UCL Language and Cognition Department Ethics, Project ID LCD-2023-06. Work undertaken at stages 1, 2 and 3 will be published in open-access peer-reviewed journal articles and presented at international scientific conferences.

**PROSPERO registration number** CRD42022367565.

## STRENGTHS AND LIMITATIONS OF THIS STUDY

⇒ This protocol describes a rigorous three-stage study, incorporating the current literature, views of people with primary progressive phasia aphasia (PPA), their family members and international, interdisciplinary researchers to identify a core outcome set for PPA.
⇒ Collaborating research sites have been identified across 15 different countries worldwide.
⇒ Translation of consensus materials from and to English will be undertaken by the research collaborators.
⇒ Consensus groups will be facilitated by speech and language researchers.
⇒ The patient and public involvement advisory group will support the analysis of results.

## BACKGROUND

Primary progressive aphasia (PPA) is a term used to describe a group of speech and language-led dementias that advance inexorably over time.[1–3] Three major PPA syndromes have been identified. Semantic variant PPA (svPPA) presents with difficulties in producing and understanding words and tends to be associated with frontotemporal lobar degeneration. Non-fluent variant (nfvPPA) is characterised by a motor speech disorder called speech apraxia, which presents as groping effortful speech, and/or difficulties in using grammar (agrammatism) and also tends to be associated with frontotemporal lobar degeneration. PPA apraxia of speech (PPA AOS) describes a 'purer' form of nfvPPA and can be further fractioned into phonetic or prosodic subtypes, but is generally considered part of the broader nfvPPA syndrome. Finally, logopenic variant PPA (lvPPA) results in difficulties in word retrieval and phonological working memory and tends to be associated with Alzheimer's pathology.[3]

There is currently no disease-modifying treatment for any form of PPA and symptomatic pharmacological therapies have also not shown evidence of effectiveness across all PPA variants. Speech and language therapists, psychologists and neuroscientists across the world have, however, developed multiple tailored interventions for people with PPA.[4–8] Yet, research examining the effectiveness of these interventions has generally focused on case studies and cohorts with small sample

sizes. Only one randomised controlled trial has been reported in the intervention literature to date.[9] Importantly, there has also been huge variability in outcome measures used across all these studies.[4 8]

Given the rarity and clinical heterogeneity of this disease, being able to group data would strengthen outcomes and knowledge about the generalisability of interventions for people with PPA. Similarly, comparing data can enhance evidence-based clinical decision making and service development.[10] Ensuring that measures are meaningful to key stakeholders, including people with the disease, their families and relevant healthcare professionals will be vital to this process.[11] The huge variability in outcomes used within the PPA intervention literature places limitations on the aggregation of outcome data, comparisons across studies and ultimately the delivery of appropriate clinical services.

A core outcome set (COS) is an agreed set of outcomes that are measured and reported in intervention studies related to a particular health condition.[11] The main reason to develop a COS is to allow for comparison across similar trials, standardise reporting to reduce selective and/or biased reporting and make results more relevant.[12] There has been previous work undertaken to develop a COS for use in the evaluation of non-pharmacological interventions for all-cause dementia which identified a long list of 54 items.[13 14] Although this generic list included communication, the list was not salient to interventions with similar objectives (it encompassed a wide variety) and did not reflect the views of stakeholders internationally.[13 14] The authors of the dementia COS also acknowledged that the constructs identified in this work are broad and overlapping and therefore lacking specificity[15] in terms of the measures to be used.

Work has been undertaken to identify a COS for post-stroke aphasia. The Research Outcome Measurement in Aphasia (ROMA)-COS identified five essential outcome constructs and appropriate measurement instruments that address each domain,[16 17] including language, communication, patient-reported satisfaction with treatment and impact of treatment, emotional well-being and quality of life. Importantly, the scope of the ROMA-COS focused on rehabilitation of non-progressive aphasia. Given people with PPA are living with a progressive disease, intervention outcomes will be different from those living with an acute onset and potentially improving aphasia.[18] Additionally, given the heterogeneity of PPA, a COS should include consideration of the value of different outcomes for different PPA syndromes.[19] We have no roadmap at present for determining or evaluating intervention outcomes in PPA and it presents radically different challenges to stroke aphasia (the current standard for aphasia interventions)—both due to its intrinsically progressive nature and also because it entails significant issues with non-verbal cognition and behaviour over the course of the illness that interacts with communication function—thus, there is a fundamental need to reorient researchers and clinicians to PPA.[18–20] In summary, there is a need for

a specific COS, that details key measures addressing the needs of people with PPA.

There is also an urgent need to improve access to care and support in PPA for people from socioeconomic, linguistic and culturally diverse backgrounds. It is anticipated that if the constructs identified are in any way similar to the ROMA-COS, they will include constructs relating to participation, capabilities and well-being that traverse linguistic contexts. Therefore, including people with PPA and their families from a diverse range of linguistic and cultural backgrounds in the development of the COS is essential to capture the voices of people from these underserved communities. To address potential barriers to the implementation of measures, incorporating an international and cross-disciplinary stakeholder perspective will promote the uptake of the COS across the research community internationally. This, in turn, will promote opportunities for future international collaborations.

The aim of this international cross-disciplinary collaboration is to develop a core set of outcome measures for researchers in the field of PPA interventions. Identifying 'what' and 'how' best to measure outcomes will inform future developments in PPA intervention research, as well as improve the impact of the work being undertaken. This, in turn, will benefit individuals with PPA by increasing access to evidence-based interventions. Thus, the objectives of this study are:

► To extract the speech, language and communication measures used to examine the effectiveness of interventions for PPA to date in the research literature.
► To identify the most important outcomes to key stakeholders, including people with PPA, their families and speech and language therapists.
► To identify outcomes that address the needs of people with different PPA variants.
► To achieve consensus on a COS for PPA (the PPA-COS) among a range of research disciplines undertaking intervention studies for people with PPA.

Scope:

1. The health condition and population covered by this COS:
   The PPA-COS covers people with PPA (inclusive of PPAOS), as defined by the current internationally accepted diagnostic criteria.[1]
2. The interventions covered by this COS:
   The PPA-COS covers intervention research that aims to affect all domains of speech, language and communication, including communication-related quality of life and well-being, for people living with PPA and their communication partners.
3. The settings covered by this COS:
   The setting covered by this COS-PPA covers intervention research and clinical delivery of interventions to people with PPA internationally, with the goal of capturing key stakeholders across all major WHO regions of the world[21] who speak and work with people with PPA in a variety of languages. We anticipate that there

may be issues with measures that have not yet been translated into languages other than English and will consider the implications of this during this process.

## METHODS

The 11 minimum standards outlined in the COS-Standards for Development Recommendations (COS-STAD)[12] informed the development of this protocol to ensure the scope, stakeholders and consensus processes were all addressed in line with these recommendations. The COS-Standardised Protocol (COS-STAP)[22] Protocol Items consisting of a checklist of 13 items and accompanying COS-STAP Explanation and Elaboration (E+E)[22] document were also consulted to ensure the COS-PPA development plans, as well as stakeholder involvement and consensus processes, were adequately addressed (see online supplemental file 2 information for completed COS-STAP reporting checklist). The COS-PPA was registered on the COMET website in March 2021: https://www.comet-initiative.org/Studies/Details/1871. In addition, the stage 1 systematic review was preregistered on PROSPERO in December 2022: CRD42022367565.

### Stakeholders
#### Members of the research team
##### Patient and public involvement (PPI)
In line with guidance from the National Institute for Health and Care Research (NIHR), PPI work encourages researchers to do research with people 'with' rather than 'to' people with lived experience.[23] The development of this study proposal was informed by PPI work undertaken during the Better Conversations with PPA (BCPPA) randomised controlled pilot study.[9] PPI collaborators identified that measurement tools needed to reflect what was important to people living with PPA.[24] The current COS-PPA study is overseen by the BCPPA and Other Rare Diseases Study PPI group. Members of this PPI group were invited from the Rare Dementia Support PPA Support Group (https://www.raredementiasupport.org/) using purposeful recruitment strategies. The BCPPA and other rare disease study PPI group are responsible for providing guidance and advice on three work packages being undertaken throughout the first author's current NIHR-funded fellowship award. The lead author has, for example, recently been working with this group to explore the limitations of the PPA-COS Stage 2 study design and future dissemination plans for the work that has informed this protocol paper. The group meets four times annually and comprises five couples where one person in each couple has PPA. The group comprises three couples where one person has mild/moderate nfvPPA, two with mild/moderate lvPPA and their spousal partners, as well as one paid carer. Further individual PPI work is ongoing with people with svPPA and those in later stages of PPA who preferred individual over group PPI contributions. All people with PPA and their partners involved in PPI activities are offered an honorarium and reimbursed for travel costs, in line with NIHR guidance.[23] Although PPI representatives reflect people with all major subtypes of PPA, there is limited socio-economic and ethnic diversity within the group, an issue the group plans to explore more broadly in the context of people with PPA accessing support services. PPI work will also be undertaken with clinical speech and language therapists within the UK PPA speech and language therapist network: a group of clinical speech and language therapists with more than 80 members. Presentations and surveys will be used to consult speech and language therapists for their opinions on the most important outcomes for people with PPA.

Current collaborators who are members of existing research networks such as the International PPA Speech and Language Therapy/Pathology Network (https://speechtherapyppa.com/)and the Collaboration of Aphasia Trialists (https://www.aphasiatrials.org/) will be contacted regarding the study, and new collaborators across WHO world regions and who speak and work with people with PPA in a variety of languages will be identified. These networks will be asked via their executive committees and leads to disseminate information about the study to all relevant working groups. Collaborators will also be identified through existing links with psychology, neuropsychology, occupational therapy and neurology disciplines, contacting authors from these disciplines who have published intervention studies and through larger networks such as the International Society on Frontotemporal Dementia (https://isftd.org) and the Include Network: For Global Equity in Language-Based Brain Health Research (https://www.gbhi.org/news-publications/include-network-global-equity-language-based-brain-health-research).

##### The study steering group
In addition to the main research team (see author list), the study will be overseen by the BCPPA and other rare diseases steering group, comprising the lead author, AV, chaired by Dr Paul Conroy, attended by Dr Cath Mummery, Consultant Neurologist, Rosemary Townsend, Speech and Language Therapist in a third sector organisation, Hannah Luff, Speech and Language Therapist in an NHS trust, Phillip Robinson, carer of a person with PPA and Samantha Stern, Neuropsychologist.

#### Study participants
Throughout this study, key stakeholders will be involved as participants. In line with previous work undertaken in the field of dementia research,[14 15] three key stakeholder groups have been identified:

People living with PPA: People diagnosed with PPA of any variant.

Family care partners: Unpaid family care partner for a person living with PPA, may include spousal, partners, siblings, adult children, or close friends.

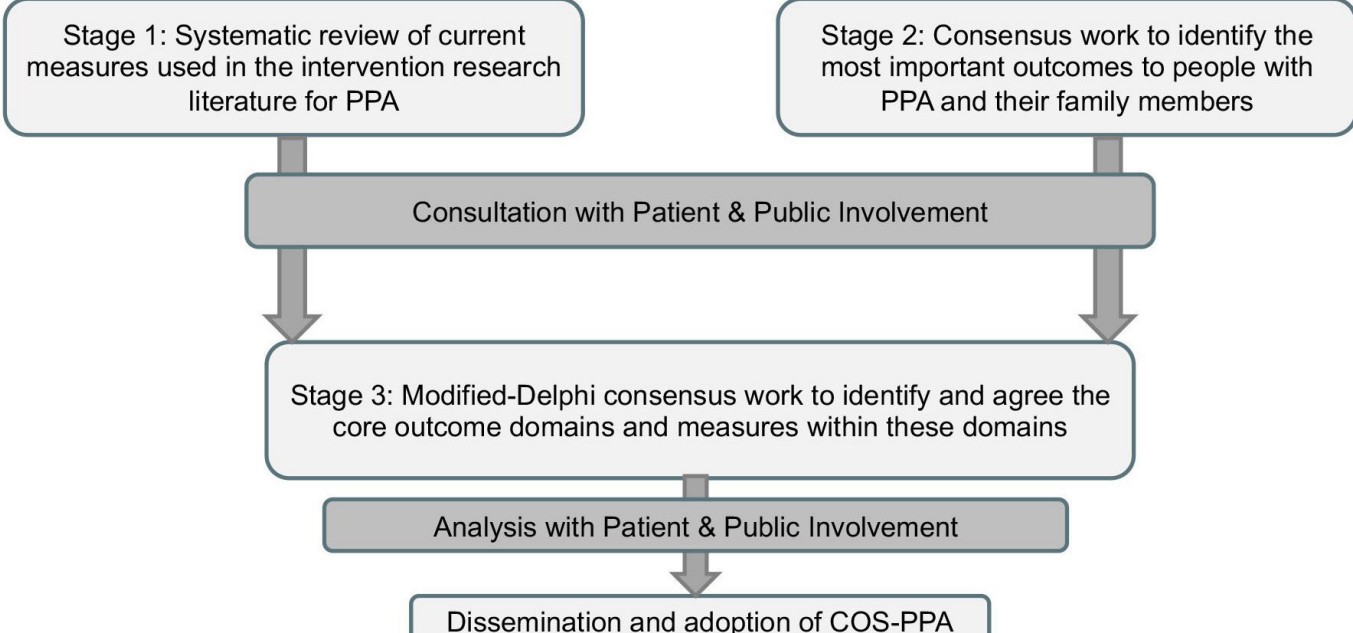

**Figure 1** Workflow for core outcome set-primary progressive aphasia (COS-PPA).

Researchers: Researchers from all backgrounds including speech and language, neurology, psychology and all those who identify as working currently or recently, undertaking PPA-related research (ie, as denoted by being a lead or co-author on dementia-related peer-reviewed publications or involvement in current dementia-related research).

### Design

The COS-PPA will be developed over three stages: stage 1—a systematic review of current measures described in the intervention research literature for PPA; stage 2—consensus work to identify the most important outcomes for people with PPA and their family members; and stage 3—a modified Delphi consensus study with researchers working in the field of PPA intervention research to agree with the core outcomes and measurement set.

Importantly, the work will endeavour to reflect an international perspective to ensure it is representative of the needs of people with PPA and their families across the world. Given some measurement tools may not be available in all languages, this study may provide a template for prioritising tool development in languages not covered by existing measures.

This approach follows the COMET handbook,[10] a guide for developing core sets of outcomes and measurements. Figure 1 provides an overview of the workflow for the COS-PPA study.

### Stage 1: A systematic review of current measures used in the intervention research literature for PPA

The aim of this systematic review is to examine the speech, language and communication measures used to examine the effectiveness of interventions for PPA (behavioural, pharmacological or neuromodulation) to date in order to:

► Identify the constructs that are measured, and how they align with the WHO International Classification of Functioning, Disability and Health (ICF) domains.
► Identify the relevance of these measures to the PPA variants; svPPA, lvPPA, nfvPPA and atypical PPA.
► Explore the psychometric properties of the measures using the COnsensus-based Standards for the selection of health Measurement INstruments (COSMIN) guidance[24] on reviewing outcome measures.

### Procedures

This systematic review will replicate and update a recent review undertaken in the field of PPA interventions,[25] employing the same search strategy but expanding this to include non-pharmacological interventions. Studies will be included that: (a) describe original research (any design), (b) are published in a peer-reviewed journal (exclusive of conference abstracts), (c) investigate behavioural, pharmacological or neuromodulation treatment for speech and/or language, (d) have been conducted with one or more persons with a diagnosis of PPA and (e) report treatment outcomes for at least one individual. Nine databases will be searched: Medline; CINAHL; (all via EBSCOhost); Embase; PsycInfo; ComDisDome; Scopus and Web of Science. Papers/measures published in languages other than English will not be excluded from the study. All outcome measures extracted will be examined by the lead author AV and those that meet the above-mentioned inclusion criteria will be included in the study. Titles and abstracts of identified articles will be reviewed by the lead author and independently reviewed by a second author (CJDH) and

assessed for inclusion or exclusion. All included articles will then undergo a second round of full-text screening by AV and CJDH independently. Any discrepancies in ratings will be discussed until an agreement is reached by AV and CJDH.

### Data extraction and analysis

All primary and secondary outcome measures of speech, language and communication reported in the final list of studies will be extracted and documented in a spreadsheet. Each measure will be considered in terms of the constructs it examines and how these align with the WHO ICF domains of impairment, activity, participation, environmental factors (such as the impact of health conditions on communication partners) and communication-related quality of life.[26] Data will be extracted from each article with regard to which PPA variant measures were used (lvPPA, svPPA, nfvPPA or no specific PPA variant, taking into account current[1] and previous PPA classification criteria[27 28]). Finally, in line with the COSMIN guidance[29] on reviewing outcome measures, data will be sought and extracted on content validity, the internal structure (ie, structural validity and internal consistency, and/or item response theory/Rasch model fit); and where applicable the remaining measurement properties (ie, reliability, measurement error, hypotheses testing, cross-cultural validity, criterion validity and responsiveness). The full protocol for this review was registered on PROSPERO in December 2022: CRD42022367565.

### Stage 2: Consensus work to identify the most important outcomes to people with PPA and their families

The important intervention outcomes for people with PPA and their families will be identified using group consensus methods. A Nominal Group Technique (NGT) protocol previously developed to meet the needs of people with stroke aphasia by co-author Dr Sarah Wallace[30] will be modified for people with PPA.

### Collaborating research sites

There is growing concern about the relevance of some COS to research in low-income and middle-income countries since participation in COS studies in those areas has been limited.[12] Therefore, to ensure representation from all WHO regions of the world and ensure representation from low-income and middle-income countries, researchers and research institutions have been approached to invite them to participate. Members of the CATs and the International PPA Speech and Language Therapy/Pathology network have agreed to participate as collaborators in the study. Collaborators across 14 countries have agreed to participate and have ethical approval including the UK, Chile, Canada, Australia, India, Germany, Portugal, Italy, Israel, Spain, USA, Netherlands, Norway and Brazil. Having agreed to participate, a manual and slide deck will be shared with collaborators instructing them on the methodology of how to run the groups (remotely where required by COVID-19

restrictions) in their respective countries. Collaborators have been asked to seek ethical approval or equivalent from their relevant institution and to translate materials where required (see ethical approvals listed).

### Recruitment of participants

Collaborators will approach people with PPA and their families via networks known to them in each country and invite them, via email, to participate in a one-off group meeting. Participants will complete consent procedures required by each collaborator's institution.

### Procedures

Collaborators will collate demographic data from participants on age, sex, type of PPA (including nfvPPA, svPPA, lvPPA and no specific variant of PPA diagnosed) and time since diagnosis. We anticipate this will include both mildly and more severely affected people with PPA. Meetings will be held either online, via video conferencing, or in person, depending on the current COVID-19 restrictions and ethical approval guidance within the relevant country. Participants with PPA will attend one meeting and family members will attend a different meeting. Participants with PPA will be asked, 'What would you most like to change about your communication and the way PPA affects your life?'. Family members will be asked, 'What would you most like to change about your family member's communication and the way PPA affects your life?'. Following NGT[31 32] methods, the participants will be invited to generate a list of items in response to the question, which will be shared in a round-robin style with the remaining group members. Having collated a list of ideas, participants will be asked to identify their top three items in the order of priority. A copy of the study manual is available in the online supplemental file 1.

### Analysis

Demographic data will be anonymised, and descriptive statistics comprising mean values will be calculated. The group facilitator (the collaborator in each country) will weigh each answer (top items will be weighted with a 3, second with a 2 and third with a 1), and aggregate individual scores to produce a final list of results, identifying the top three rated items for each group (people with PPA and their families) in each country. Anonymised results collected from different countries will be shared with the lead author (AV), who will aggregate items in line with NGT methodology[31 32] and produce an overall list of results and demographic data. Results will consider both an overall PPA outcomes list of results and those that align with particular PPA variants.

### Ethical approval

Ethical has been sought by each collaborator at their participating institutions, and participants will be consented based on local guidance. The stage 2 UK collaboration is being undertaken as part of the Rare Dementia Support (RDS) Impact Study which received approval from the UCL Research Ethics Committee



(8545/004: Rare Dementia Support Impact Study). All consent sessions will be video recorded, in line with the approved procedure outlined in the RDS Impact study protocol (Brotherhood *et al*, 2020).[33] For the collaboration with Dr Carolina Mendez in Chile, ethical approval has been granted by the Pontificia Universidad Catolixa de Chile Ethics Committee, ID no 190 510 002. For the collaboration with Dr Regina Jokel in Canada, ethical approval has been granted by Baycrest, Research Ethics Board REB 22-37. For the collaboration with Dr Jade Cartwright and Dr Cathy Taylor-Rubin in Australia, ethical approval has been granted by Southeastern Sydney Local Health District HREC 2022/ETH02740. For the collaboration with Dr Avanthi Paplikar in India, ethical approval has been granted by the Bangalore Speech and Hearing Research Foundation. For the collaboration with Prof Marcus Meinzer, Anna Rysop and Nina Unger in Germany ethical approval has been granted by Griefswald University Ethics Committee, Germany, Reference BB 130/22. For the collaboration with De Ines Cadorio in Portugal, ethical approval has been granted by Ethics Committee of the University Fernando Pessoa Prot n. 50 /C.E> del 28February 2022. For the collaboration with Dr Petronilla Battista in Italy, ethical approval was granted by the Ethics Committee of the IRCCS Giovanni Paolo II Bari, Prot. n. 80/CE Maugeri on 17 February 2022. For the collaboration with Dr Adi Lifshitz Ben Basat and Hagit Bar-Zeev in Israel, ethical approval was granted by the Ariel University Ethics Board, Israel. For the collaboration with Dr Maya Henry and Carly Milanski in America, ethical approval was given by the Office of Research Support and Compliance and the University of Texas at Austin's Institutional Review Board IRB ID STUDY00000717-MOD06. For the collaboration with Dr Sandra Weilart, Dr Lize Jiskoot, Janna Vanegmond, Heleen Hendriksen and Antoinette Keulen in the Netherlands, ethical approval was granted by Amsterdam UMC under the number 2023.0098. For the collaboration with Dr Monica Norvik in Norway, ethical approval was granted by the Norwegian Agency for Shared Services in Education and Research (SIKT) ref: 865 145. For collaboration with Maria Isabel d'Avila Freitas in Brazil, ethical approval was granted by Hospital of Clinics—Faculty of Medicine—Uni of Sao Paulo (USP) ref: 4.142.664.

Data collected from both stages 1 and 2 will be presented to the project PPI group for analysis and discussion to agree on the key constructs that should inform discussion at the final consensus group. Given that several members of the PPI group have communication difficulties, information will be presented in an accessible format, using images and aphasia-accessible written and spoken language. Gaps in the data and reasons for this will be considered and will inform prioritisation of information for the final consensus meeting.

### Stage 3: COS-PPA consensus work to identify and agree core outcomes and measurement sets

Consensus work to agree a final COS-PPA with the team of international cross-disciplinary researchers will be undertaken using Delphi consensus methods.[31] This will include identifying and agreeing the core outcome measurement instruments for each of key constructs identified in stages 1 and 2. Stage 3 work has been granted ethical approved by UCL Language and Cognition Department Ethics Committee, Project ID LCD-2023-06.

#### Participants

Participants will be recruited via international networks of researchers working in the field of PPA intervention research. Given researchers in this field may include speech and language researchers, neuropsychology, psychology, neurology, occupational therapy and other groups, it will be essential to explore networks within and outside the International PPA Speech and Language Therapy/Pathology network and CATS. Researchers and authors of studies identified in the stage 1 systematic review will be contacted, in addition to contacting researchers via additional networks including the International Society of Fronto-Temporal Dementia (ISFTD). The authors will purposefully aim to recruit researchers across a range of countries, without overly biasing representation from one country.

#### Procedures

Emails will be sent inviting researchers to participate in the study. Should they agree by the return of email, they will be sent a link to complete an online consent form and brief survey collecting demographic information (including affiliation, professional background, research interests, country of current work, qualification, number of PPA participants seen as part of the research, languages in which research undertaken) and availability to attend a meeting hosted on a video conferencing platform.

Prior to attending video conferencing meetings, researchers will be asked to complete an online vote rating the importance of outcome constructs (selected based on data collated in stage 2 of the study). Researchers will rate each construct on a scale of importance, with 9 being the most important and 1 being the least important. Rankings of 7–9 indicate critical importance, 4–6 outcomes are important but not critical, while ratings of 1–3 are of limited importance using the Grading of Recommendations Assessment, Development and Evaluations (GRADE) scale.[34] Researchers will be invited to put other constructs forward that they think are important but had not been included in the survey. They will also be asked about which constructs require different measurements for individual PPA variants and which do not. These results will be combined with the stage 2 NGT work (whereby stage 2 ratings will be converted to weighted scores 1-9 outlined above), and a mean rating provided. Constructs that receive a rating of 6-9 will be taken forward to the next stage.

Only researchers who have completed the online ratings will be invited to participate in online meetings. At least two online meetings, hosted on Zoom, will be held to capture researchers across different zones. Meetings will be facilitated by an independent facilitator, not affiliated with research in this field. For each construct, researchers will be presented with measures identified in the stage 1 systematic review that measure this construct. Researchers will be provided with a description of each measure, including its psychometric properties and languages in which it is available, and will then be asked to vote on which measures they feel best address the construct. Measures will not be excluded if there is no psychometric data available. Researchers will also be asked to put forward any alternative measures. Results of the first round of voting, including any new suggestions, will be disseminated to the researchers via email. They will then be invited to re-vote, to identify which measure would be best as a core outcome measure for each construct.

### Analysis

Demographic information will be analysed using descriptive statistics. Anonymised voting data collected in the modified Delphi consensus study will be aggregated and data italicised for presentation.

### Dissemination

Work undertaken at stages 1, 2 and 3 will be published in open-access peer-reviewed journal articles and presented at international scientific conferences. The results will additionally be published in clinical practice magazines in the UK, the Royal College of Speech and Language Therapy Bulletin magazine and Practical Neurology, and equivalent non-UK practice outlets identified by collaborators in relevant participating countries. The work will also be submitted to the WHO's Global Observatory Knowledge Exchange Platform,[35] a platform designed specifically to share knowledge about dementia research in response to the Global Action Plan on the Public Health Response to Dementia 2017–2025.[36]

Prior to publication, findings from the stage 2 work to identify the most important outcomes for people with PPA and their families will be shared with participating international collaborators in the form of a draft manuscript. They will be invited to be co-authors and to comment on the manuscript prior to submission to a peer-reviewed journal. Additionally, it will be suggested that they share co-produced accessible materials with local participants.

Consensus work undertaken at stage 3 will be shared by email with participants. Researchers participating in the study will be invited, at the outset, to collaborate on a peer-reviewed manuscript, as co-authors. Having remained involved in the consensus process, researchers will be reminded of this and invited to collaborate and comment on a draft manuscript. As co-authors on the paper, they will assist in the dissemination of the results

as well as receiving continuing updates on the research study.

The PPI contributors and the study steering group will be acknowledged in all submitted manuscripts and scientific conference presentations.

### DISCUSSION

This protocol describes the development of a COS for PPA (COS-PPA) following three stages to identify the current measures used in the intervention research literature, what is important to people with PPA and their families internationally and finally a modified-Delphi consensus exercise to agree to the COS. There is currently no published COS for PPA, and existing COSs for dementia[14 15] and non-progressive aphasia[16 17] do not address the unique and specific needs of this group. The current research literature on PPA interventions uses a variety of outcomes,[8] limiting comparability across studies and does not address what is important to people with PPA and their families. Identifying measures that reflect the needs of people with all PPA variants will be challenging. It is essential that we understand the priorities of this seldom-heard group of people with speech, language and communication difficulties and that we embed these in our research design. Using a COS could allow for the combination of datasets, which in the current setting of smaller studies would increase the power of the intervention studies and increase our understanding of the efficacy of interventions for PPA. Given our collaborations internationally, the intention is for this COS to be used internationally, thus the dissemination plan includes both publications in peer-reviewed journals and international dissemination platforms. We anticipate that there will be a few measures available that have been translated into different languages. In fact, it is likely that there may be no suitable measures to address all the identified constructs that are validated or developed for use with people with PPA. This highlights the potential for this study to inform and help prioritise more research to explore how PPA presents differently across languages, in both monolingual and multilingual speakers. Our ambition is to support the development of new measures for people with PPA, and their translation and validation across different languages. Given the differences in languages, it is anticipated that these measures may not necessarily focus on linguistic performance but on patient-reported outcome measures. As an example, a communication function such as coherent and connected propositional speech is key to effective communication in English, Chinese or Turkish (etc) even though the specific linguistic vehicles may vary widely between languages. This in turn will potentially have far-reaching implications beyond the scope of PPA, extending to people living with other dementia types, who also have speech, language and

communication needs such as Posterior Cortical Atrophy,[37] Frontotemporal Dementia, Young Onset Alzheimer's Disease, Typical Alzheimer's Disease and Lewy Body Dementia.[38]

## Trial status

Stage 1 Systematic review searches March 2023, data extraction May-August 2023.

Stage 2 Nominal groups with people with PPA July 2021 to October 2023

Stage 3 Modified Delphi Consensus methods November 2023 to December 2023

## Ethical approval

All aspects of the study will be conducted according to the Declaration of Helsinki.

The stage 2 UK collaboration is being undertaken as part of the Rare Dementia Support (RDS) Impact Study which received approval from the UCL Research Ethics Committee (8545/004: Rare Dementia Support Impact Study). All consent sessions will be video recorded, in line with the approved procedure outlined in the RDS Impact study protocol (Brotherhood *et al.* 2020). For the collaboration with Dr Carolina Mendez in Chile, ethical approval has been granted by the Pontificia Universidad Catolixa de Chile Ethics Committee, ID no 190 510 002. For the collaboration with Dr Regina Jokel in Canada, ethical approval has been granted by Baycrest, Research Ethics Board REB 22-37. For the collaboration with Dr Jade Cartwright and Dr Cathy Taylor-Rubin in Australia, ethical approval has been granted by Southeastern Sydney Local Health District HREC 2022/ETH02740. For the collaboration with Dr Avanthi Paplikar in India, ethical approval has been granted by the Bangalore Speech and Hearing Research Foundation. For the collaboration with Prof Marcus Meinzer, Anna Rysop and Nina Unger in Germany, ethical approval has been granted by Griefswald University Ethics Committee, Germany, Reference BB 130/22. For the collaboration with De Ines Cadorio in Portugal, ethical approval has been granted by the Ethics Committee of the University Fernando Pessoa Prot n. 50 /C.E> del 28 February 2022. For the collaboration with Dr Petronilla Battista in Italy, ethical approval was granted by the Ethics Committee of the IRCCS Giovanni Paolo II Bari, Prot. n. 80/CE Maugeri on 17 February 2022. For the collaboration with Dr Adi Lifshitz Ben Basat and Hagit Bar-Zeev in Israel, ethical approval was granted by the Ariel University Ethics Board, Israel. For the collaboration with Dr Maya Henry and Carly Milanski in America, ethical approval was given by the Office of Research Support and Compliance and the University of Texas at Austin's Institutional Review Board IRB ID STUDY00000717-MOD06. For the collaboration with Dr Sandra Weilart, Dr Lize Jiskoot, Janna Vanegmond, Heleen Hendriksen and Antoinette Keulen in the Netherlands, ethical approval was granted by Amsterdam UMC under the number 2023.0098. For the collaboration with Dr Monica Norvik in Norway, ethical approval was granted by the Norwegian Agency for Shared Services in Education and Research (SIKT) ref: 865 145. For collaboration with Maria Isabel d'Avila Freitas in Brazil, ethical approval was granted by Hospital of Clinics—Faculty of Medicine—Uni of Sao Paulo (USP) ref: 4.142.664.

Stage 3 work has been granted ethical approval by the Chairs of UCL Language and Cognition Department Ethics, Project ID LCD-2023-06. All participants involved will be asked to complete a written form of consent before participation in the Delphi survey.

**Acknowledgements** The authors would like to acknowledge the invaluable assistance and advice of all the PPI contributors and the study steering group (Dr Paul Conroy, Dr Cath Mummery, Rosemary Townsend, Hannah Luff, Phillip Robinson and Samantha Stern). They would also like to thank all collaborators across all collaborating countries; Dr Carolina Mendez in Chile, Dr Regina Jokel in Canada, Dr Jade Cartwright and Dr Cathy Taylor-Rubin in Australia, Dr Avanthi Paplikar in India, Prof Marcus Meinzer, Anna Rysop and Nina Unger in, Dr Ines Cadorio in Portugal, Dr Petronilla Battista in Italy, Dr Adi Lifshitz Ben Basat and Hagit Bar-Zeev in Israel, Dr Maya Henry and Carly Milanski in America, Dr Sandra Weilart, Dr Lize Jiskoot, Janna Vanegmond, Heleen Hendriksen and Antoinette Keulen in the Netherlands, Dr Monica Norvik in Norway, Maria Isabel d'Avila Freitas in Brazil, Dr Ibrahim Can Yasa in Turkey and Antoine Renard in France.

**Contributors** AV is the chief investigator; she conceived the study and led the proposal and protocol development. CJDH contributed to the study design and development of the proposal. SJW and DAC contributed to the study design. RV, MLH and JDW contributed to finalising the proposal. All authors have contributed to the manuscript preparation.

**Funding** This study was initially funded by a UCL Grand Challenge awarded to AV and CH in 2020, and consequently by an NIHR Advanced Fellowship NIHR302240 to AV. SJJW is funded by a National Health and Medical Research Council (NHMRC) Investigator Grant (1175821). JW has received grant support from the Alzheimer's Society, Alzheimer's Research UK, the Royal National Institute for Deaf People (Discovery Grant G105_WARREN), the National Institute for Health Research University College London Hospitals Biomedical Research Centre and the National Brain Appeal (Frontotemporal Dementia Research Studentship in Memory of David Blechner).

**Competing interests** None declared.

**Patient and public involvement** Patients and/or the public were involved in the design, or conduct, or reporting, or dissemination plans of this research. Refer to the Methods section for further details.

**Patient consent for publication** Not applicable.

**Provenance and peer review** Not commissioned; externally peer reviewed.

**ORCID iDs**
Anna Volkmer http://orcid.org/0000-0002-4149-409X
Jason D Warren http://orcid.org/0000-0002-5405-0826
Sarah J Wallace http://orcid.org/0000-0002-0600-9343
Chris JD Hardy http://orcid.org/0000-0002-4900-6492

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
