## [Reviewer comments · BMJ Open]

ARTICLE DETAILS

TITLE (PROVISIONAL)	COS- PPA: Protocol to develop a core outcome set for Primary Progressive Aphasia
AUTHORS	Volkmer, Anna; Copland, David; Henry, Maya L.; Warren, Jason; Varley, Rosemary; Wallace, Sarah J.; Hardy, Chris

VERSION 1 – REVIEW

REVIEWER	Mitchell, Claire Manchester University
REVIEW RETURNED	13-Dec-2023

GENERAL COMMENTS	Thank you for submitting this protocol of your planned work. The international element to your work is impressive and will have benefits for the population globally which is to be commended. This planned protocol is clearly explained and I make some minor suggestions that might improve the paper for the reader. Abstract page 2 Background line 10 – Not clear how this term ‘disease-modifying treatments’ is different to ‘symptomatic interventions’ - needs more explanation or to be described in terms the reader might understand. It seems to be comparing treatment for the dementia as opposed to treatment for the PPA specifically – needs to be clarified. You go on to explain this well in the background but when reading the abstract this needs to be made clear at this early point in the paper where you have to explain this in more concise language. 31-33 – who are the key stakeholders – only know people with PPA and family – need to know who else. Page 4 line 43-45 There is also an urgent need to improve diversity of access to care and supports in PPA, socioeconomically, linguistically and culturally. Not clear how this relates to the COS – needs more explanation or background to what you mean by this and what the connection is. Line 48,49,50 – there are other barriers though to implementing COSs though – might be helpful to give that broader overview as well. Page 8 – design This was difficult to follow due to the different participants at the different stages. Who is involved at these stages needs to be made clearer see below for suggestions.
---

	Page 10 line 22 – talks about important to key stakeholders – so who do you mean in stage 2 as the key stakeholders? Can you specify this early as not clear to reader. On page 8 line 18/19 you say this is people with PPA but page 10 stage 2 – you don't seem to mention people with PPA just low and middle income countries. Not clear who the participants or key stakeholders are at this point. Page 11 46-52 – how will this very complex information be presented so accessible to the PPI group – might be good to just state you're aware that this is a challenge. Page 11 – line 55 – stage 3 who are the participants at this stage. Do the collaborators also include PPI or people with PPA – not clear. If only researchers that needs stating. Page 13 lines 5-6. When you complete the systematic review you don't know what constructs will be identified so you talk about presenting people with measures related to each construct. You might not find any measures, these measures might not have psychometric data that warrants inclusion in a COS so this might be impossible. I think this needs to be more nuanced and more open to what you may or may not find in your syst review. You touch on this in your discussion but at this point in the paper you have to show you have prepared for this eventuality and what you might do if no papers or of poor quality.
--	---

REVIEWER	Rofes, Adria Global Brain Health Institute
REVIEW RETURNED	27-Feb-2024

GENERAL COMMENTS	I am pleased to provide feedback on the paper titled “COS- PPA: Protocol to Develop a Core Outcome Set for Primary Progressive Aphasia” submitted to BMJ Open by Anna Volkmer and colleagues. The paper underscores the necessity of establishing a specific Core Outcome Set (COS) tailored for individuals with Primary Progressive Aphasia (PPA). The authors outline a three-step approach to achieve this objective: (1) a systematic review of the literature; (2) employment of the nominal group technique; and (3) conducting a Delphi consensus study. Below are some major and minor comments: Major comments: While I acknowledge the authors' efforts, I have reservations regarding the significance of this study. I recognize the importance of developing a COS for PPA akin to that of post-stroke aphasia. However, it remains unclear why readers would find the protocol compelling without the presentation of actual study results. I am particularly interested in understanding the specifics of the COS for PPA and would appreciate a critical evaluation of its divergence from the COS for other types of aphasia. Furthermore, I would welcome insights from experts on whether COS for PPA should vary across different languages or if a Cognitive Neuropsychological approach could identify language components of particular significance.
--

Moreover, while I appreciate the relevance of publishing this protocol, I believe there is a need for more comprehensive information on PPA in languages other than English. Methodologically, I would have liked to see predictions derived from existing literature for each phase of the study. If the authors choose to pursue this direction, some of this information could be incorporated into the discussion section. As it stands, the discussion appears somewhat redundant with the introduction. Hence, it is not the most compelling section to read.

Minor comments

p3, line 14 – please change “progress” for a synonym such as “evolve”, “develop”, or “advance”.

p3, line 17 – please write “tends to be associated with frontotemporal lobar degeneration”

p3, line 23 – please write “tends to be associated with frontotemporal lobar degeneration”

p3, line 30 – please write “tends to be associated with Alzheimer’s pathology” instead of “chiefly associated...”

p4, line 12 – please use COS instead of the full word

p4, line 22 – please include a reference after “internationally”

p4, line 23 – is “the review” the current study or another study?

p4, line 29 – please consider changing “stroke aphasia” for “post-stroke aphasia”.

p4, line 45 – it is unclear why the authors do not refer to the progressive nature of the disease here. Is it the case that COS for PPA should be different than those for post-stroke aphasia, also because PPA is a neurodegenerating disease and multiple assessments with inevitably varying outcomes accrue vs post-stroke?

p4, lines 48 and 54 – please delete “international and cross-disciplinary” I am not too sure this is relevant. If the authors want to strengthen the international aspect of the study, perhaps they should join forces with colleagues in countries where English is not the vehicular language? Some of it seems to be indicated in previous paragraphs, but I am not too sure if then linguistic variation would lead the authors to a difficult rabbit hole where some of the COS suggested may not apply to languages other than English? For example, while agrammatism may be an issue in English, this may be less prominent in Chinese (no inflection), and much more prominent or differently affected in languages such as Turkish (e.g., evidentiality, agglutinative nature).

Alternatively, the authors could refer to English speaking countries, but then it is unclear why they did not join forces with colleagues in Canada, New Zealand, Ireland, etc. The international aspect is again discussed on page 10 of this article, perhaps some of that information should be included here?

p5, line 22 – I am confused by the section “scope”. What is the goal of this? Maybe this should be deleted or written as a paragraph? I see scope is mentioned in the methods, maybe this is a section that should be included there?

p6, line 12 – the COS-PPA being registered in 2021 and now aimed to be published in 2024 makes me raise an eyebrow regarding the potential quality of this study. Why did 3 years happen between assessment times? Maybe this is something worth mentioning? Also, in the website some of the co-authors of this paper are not mentioned: Maya Henry, Jason Warren, David

	Copland (?David Copeland is mentioned in the website), Rosmary Varley. Why is this? Are these two things separate? p6, line 26 - the "Better Conversations with PPA" study comes out of the blue here. This should have been mentioned in the introduction. The same holds with the "Other Rare Diseases Study PPI group" mentioned below. p6, line 27 – what are PPA collaborators? Patients, researchers, clinicians, caregivers, advocates? How are they selected? p6, line 38 – what is the "first author's current fellowship award"? p6, line 43 – I am not sure what is "this protocol paper". p6, line 60 – Up until now I had the impression that the three studies mentioned in the abstract were already conducted and would be discussed here. However, only here I understood that this is "a protocol" and therefore that there are no results provided. I am aware that the title includes the word "protocol". However, my expectation as a reader was much different. p7, line 12 – what are "collaborators" here? Are the same than those in page 6 line 27? p7, line 17 – please change "Triallists" for "Trialists". Which working group from the Collaboration of Aphasia Trialists did you contact? Did they discuss this in their regular meetings? I am aware that some groups are much more active than others. Also, the collaboration worked rather differently before and after it received funds from the European Union. p7, line 50 – why is this section highlighted in grey? p7, line 56 – how many people with will PPA enter the study? What will be their expected demographic details? Any chance that the diagnosis will be mixed? Would any people be included who had been recently diagnosed and had mild impairments or people who had severe impairments? p9, line 29 – it is unclear why the authors mention that they want to replicate a study that is not published (under review). Can you truly replicate an unpublished work? p9, line 41 – please specify what are "non-English measures". Does this refer to the language in which the papers are written, or the language in which the assessments were conducted? p10, line 7 – It is unclear why the authors do not consider the possibility that studies do not indicate PPA variant or mixed variants. Also, some studies may not discuss PPA variant à la Gorno-Tempini, but following previous classifications by Kertesz or Mesulam. Would the authors be willing to establish some kind of parallel between classifications? What to do in cases where apraxia of speech is or not considered compulsory? p10, line 60 – "Participants will complete consent procedures required by each collaborator's institution". It is unclear how many institutions will end up participating and under which ethic approval(s) will this project be performed, if at all? Perhaps the main author has an umbrella approval that is potentially valid in other countries? It is unclear how the privacy of the participants will be protected, how the documents that will be used will be collected and/or destroyed. Perhaps all information gathered will be anonymous or pseudonymized? Later on, it is indicated that the data will be shared in anonymized format to the main author. p11, line 42 – Information on ethics approval appears incomplete. Will this information be included in the eventual study?
--	--

VERSION 1 – AUTHOR RESPONSE

Reviewer: 1

1. Abstract page 2 Background line 10 – Not clear how this term ‘disease-modifying treatments’ is different to ‘symptomatic interventions’ - needs more explanation or to be described in terms the reader might understand. It seems to be comparing treatment for the dementia as opposed to treatment for the PPA specifically – needs to be clarified. You go on to explain this well in the background but when reading the abstract this needs to be made clear at this early point in the paper where you have to explain this in more concise language.

A brief definition of disease-modifying and symptomatic interventions has been added to the abstract:

“The term Primary Progressive Aphasia (PPA) describes a group of language-led dementias. Disease-modifying treatments that delay, slow or reverse progression of PPA are currently lacking, though a number of interventions to manage the symptoms of PPA, have been developed in recent years.”

2. 31-33 – who are the key stakeholders – only know people with PPA and family – need to know who else.

The abstract has been clarified to explain who is involved in the stage 2 work, see below:

“Methods: This protocol describes a three-stage study to identify a COS for Primary Progressive Aphasia interventions in research and clinical practice. The Stage 1 systematic review will identify existing speech, language and communication measures used to examine the effectiveness of interventions for PPA in the research literature. Employing a Nominal Group Technique, Stage 2 will identify the most important outcomes to people with PPA and their families. The data collected in Stages 1 and 2 will be jointly analysed with the project PPI group and will inform the Stage 3 modified Delphi consensus study to identify a core outcome measurement set for PPA amongst a range of research disciplines undertaking intervention studies for people with PPA.”

3. Page 4 line 43-45

There is also an urgent need to improve diversity of access to care and supports in PPA, socioeconomically, linguistically and culturally.

Not clear how this relates to the COS – needs more explanation or background to what you mean by this and what the connection is.

The following has been added to explain what is meant by this:

“There is also an urgent need to improve access to care and supports in PPA for people from socioeconomic, linguistically and culturally diverse backgrounds. Including people with PPA and their families from a diverse range of linguistic and cultural backgrounds in the development of the COS is essential to capture the voices of people from these underserved communities. To address potential barriers to implementation of measures, incorporating an international and cross-disciplinary stakeholder perspective will promote uptake of the COS across the research community internationally. This, in turn, will promote opportunities for future international collaboration.”

4. Line 48,49,50 – there are other barriers though to implementing COSs though – might be helpful to give that broader overview as well.

5. Page 8 – design

This was difficult to follow due to the different participants at the different stages. Who is involved at these stages needs to be made clearer see below for suggestions.

The initial description of the design has been amended to provide clarity around who the key stakeholders are at which stages:

“The COS-PPA will be developed over three stages: Stage 1) a systematic review of current measures described in the intervention research literature for PPA; Stage 2) consensus work to identify the most important outcomes for people with PPA and their family members; and Stage 3) a modified Delphi consensus study with researchers working in the field of PPA intervention research to agree the core outcomes and measurement set.

Importantly, the work will endeavour to reflect an international perspective to ensure it is representative of the needs of people with PPA and their families across the world. Given some measurement tools may not be available in all languages, this study may provide a template for prioritising tool development in languages not covered by existing measures.”

Additionally, the figure has been updated for clarity (see uploaded document)

6. Page 10 line 22 – talks about important to key stakeholders – so who do you mean in stage 2 as the key stakeholders? Can you specify this early as not clear to reader. On page 8 line 18/19 you say this is people with PPA but page 10 stage 2 – you don’t seem to mention people with PPA just low and middle income countries. Not clear who the participants or key stakeholders are at this point.

In the description of the stage 2 consensus groups use of the term “key stakeholders” has been removed and replaced with “people with PPA and their families” throughout. For clarity the subheadings have also been modified in this section to reflect:

Collaborating research sites

Recruitment of participants

Procedures

7. Page 11 46-52 – how will this very complex information be presented so accessible to the PPI group – might be good to just state you’re aware that this is a challenge.

The following sentence has been added to capture this important point, thank you:

“Given that several members of the PPI group have communication difficulties, information will be presented in an accessible format, using images and aphasia accessible written and spoken language.”

8. Page 11 – line 55 – stage 3 who are the participants at this stage. Do the collaborators also include PPI or people with PPA – not clear. If only researchers that needs stating.

This has been made clearer, and participants have been referred to as researchers throughout this section.

9. Page 13 lines 5-6. When you complete the systematic review you don’t know what constructs will be identified so you talk about presenting people with measures related to each construct. You might not find any measures, these measures might not have psychometric data that warrants inclusion in a

COS so this might be impossible. I think this needs to be more nuanced and more open to what you may or may not find in your syst review. You touch on this in your discussion but at this point in the paper you have to show you have prepared for this eventuality and what you might do if no papers or of poor quality.

Many thanks for flagging this. The following edits to this section provide a more nuanced description of how this will be considered:

Researchers will be provided with a description of each measure, including its psychometric properties and languages in which it is available, and will then be asked to vote on which measures they feel best address the construct. Measures will not be excluded if there is no psychometric data available. Researchers will also be asked to put forward any alternative measures. Results of the first round of voting, including any new suggestions will be disseminated to the researchers via email. They will then be invited to re-vote, to identify which measure would be best as a core outcome measure for each construct.

Reviewer: 2

Dr. Adria Rofes, Global Brain Health Institute

10. While I acknowledge the authors' efforts, I have reservations regarding the significance of this study. I recognize the importance of developing a COS for PPA akin to that of post-stroke aphasia. However, it remains unclear why readers would find the protocol compelling without the presentation of actual study results. I am particularly interested in understanding the specifics of the COS for PPA and would appreciate a critical evaluation of its divergence from the COS for other types of aphasia. Furthermore, I would welcome insights from experts on whether COS for PPA should vary across different languages or if a Cognitive Neuropsychological approach could identify language components of particular significance.

We acknowledge that this paper does not present any results, but in line with recommendations for the development of Core Outcome Sets (COS-Standards for Development Recommendations (COS-STAD) and COS-Standardised Protocol (COS-STAP) [21] Protocol Items) and Open Science methods this paper outlines a rigorous protocol for the COS-PPA study. Of note, the remit of BMJ Open include protocols. We are looking forward to sharing the results of the study with the journal readers as soon as we are able.

We appreciate your suggestion to strengthen the rationale for developing a COS for PPA given the current stroke aphasia COS in existence and have addressed this by adding the following to the manuscript:

“Work has been undertaken to identify a COS for stroke aphasia. The Research Outcome Measurement in Aphasia (ROMA)-COS identified five essential outcome constructs and appropriate measurement instruments that address each domain [16,17] including: language, communication, patient-reported satisfaction with treatment and impact of treatment, emotional wellbeing, and quality of life. Importantly, the scope of the ROMA-COS focused on rehabilitation of non-progressive aphasia. Given people with PPA are living with a progressive disease, intervention outcomes will be different to those living with an acute onset, and potentially improving aphasia [18]. Additionally, given the heterogeneity of PPA, a COS should include consideration of the value of different outcomes for different PPA syndromes [19]. We have no roadmap at present for determining or evaluating intervention outcomes in PPA and it presents radically different challenges to stroke aphasia (the current standard for aphasia interventions) - both due to its intrinsically progressive nature but also because it entails significant issues with nonverbal cognition and behaviour over the course of the

illness that interact with communication function- thus there is a fundamental need to reorient researchers and clinicals to PPA [18, 19, 38]. In summary, there is a need for a specific COS, that details key measures addressing the needs of people with PPA.”

11. Moreover, while I appreciate the relevance of publishing this protocol, I believe there is a need for more comprehensive information on PPA in languages other than English. Methodologically, I would have liked to see predictions derived from existing literature for each phase of the study. If the authors choose to pursue this direction, some of this information could be incorporated into the discussion section. As it stands, the discussion appears somewhat redundant with the introduction. Hence, it is not the most compelling section to read.

Whilst we are unable to provide detailed predictions in the protocol, we have also expanded the discussion to predict that this study could promote the need for research exploring both how PPA presents differently across languages, but also inform the development of future measures that focus on patient reported outcomes:

“We anticipate that there will be few measures available that have been translated into different languages. In fact, it is likely that there may be no suitable measures to address all the identified constructs and that are validated or developed for use with people with PPA. This highlights the potential for this study to inform and help prioritise more research to explore how PPA presents differently across languages, in both monolingual and multilingual speakers. Our ambition is to support the development of new measures for people with PPA, and their translation and validation across different languages. Given the differences in languages, it is anticipated that these measures may not necessarily focus on linguistic performance, but on patient reported outcome measures. As an example, a communication function such as coherent and connected propositional speech is key to effective communication in English, Chinese or Turkish (etc) even though the specific linguistic vehicles may vary widely between languages. This in turn, will potentially have far-reaching implications beyond the scope of PPA, extending to people living with other dementia types, who also have speech, language, and communication needs such as Posterior Cortical Atrophy [34], Frontotemporal Dementia, Young Onset Alzheimer’s Disease, Typical Alzheimer’s Disease and Lewy Body Dementia [35].”

12. Minor comments

p3, line 14 – please change “progress” for a synonym such as “evolve”, “develop”, or “advance”.

p3, line 17 – please write “tends to be associated with frontotemporal lobar degeneration”

p3, line 23 – please write “tends to be associated with frontotemporal lobar degeneration”

p3, line 30 – please write “tends to be associated with Alzheimer’s pathology” instead of “chiefly associated...”

p4, line 12 – please use COS instead of the full word

p4, line 22 – please include a reference after “internationally”

p4, line 23 – is “the review” the current study or another study?

p4, line 29 – please consider changing “stroke aphasia” for “post-stroke aphasia”.

The above points have been changed in accordance with the reviewers’ suggestions.

13. p4, line 45 – it is unclear why the authors do not refer to the progressive nature of the disease here. Is it the case that COS for PPA should be different than those for post-stroke aphasia, also because PPA is a neurodegenerating disease and multiple assessments with inevitably varying outcomes accrue vs post- stroke?

Please see comments above, in response to point 10 which we hope address this concern.

14. p4, lines 48 and 54 – please delete “international and cross-disciplinary” I am not too sure this is relevant. If the authors want to strengthen the international aspect of the study, perhaps they should join forces with colleagues in countries where English is not the vehicular language? Some of it seems to be indicated in previous paragraphs, but I am not too sure if then linguistic variation would

lead the authors to a difficult rabbit hole where some of the COS suggested may not apply to languages other than English? For example, while agrammatism may be an issue in English, this may be less prominent in Chinese (no inflection), and much more prominent or differently affected in languages such as Turkish (e.g., evidentiality, agglutinative nature). Alternatively, the authors could refer to English speaking countries, but then it is unclear why they did not join forces with colleagues in Canada, New Zealand, Ireland, etc. The international aspect is again discussed in page 10 of this article, perhaps some of that information should be included here?

We believe it is extremely important for the COS to include a strong cross-linguistic focus, given that we do not anticipate identifying only linguistic outcome constructs. The additions have been included in the introduction to make this clearer:

“There is also an urgent need to improve access to care and supports in PPA for people from socioeconomic, linguistically and culturally diverse backgrounds. It is anticipated that if constructs identified are in anyway similar to the ROMA-COS, they will include constructs relating to participation, capabilities and wellbeing that traverse linguistic contexts. Therefore, including people with PPA and their families from a diverse range of linguistic and cultural backgrounds in the development of the COS is essential to capture the voices of people from these underserved communities. To address potential barriers to implementation of measures, incorporating an international and cross-disciplinary stakeholder perspective will promote uptake of the COS across the research community internationally. This, in turn, will promote opportunities for future international collaborations.”

15. p5, line 22 – I am confused by the section “scope”. What is the goal of this? Maybe this should be deleted or written as a paragraph? I see scope is mentioned in the methods, maybe this is a section that should be included there?

The protocol has been written in accordance with the relevant guidelines identified in point 10 which recommend this section (scope) be included here. We hope this addresses the reviewers concerns and with this in mind we have made no amendments to this section.

16. p6, line 12 – the COS-PPA being registered in 2021 and now aimed to be published in 2024 makes me raise an eyebrow regarding the potential quality of this study. Why did 3 years happen between assessment times? Maybe this is something worth mentioning? Also, in the website some of the co- authors of this paper are not mentioned: Maya Henry, Jason Warren, David Copland (?David Copeland is mentioned in the website), Rosemary Varley. Why is this? Are these two things separate?

The project was registered prior to any work on the project being undertaken, in line with Open Science recommendations. Since this time, funding has been secured and this protocol has been under review for more than 8-months. We hope this addresses the reviewers concerns about the project timeline.

17. p6, line 26 - the “Better Conversations with PPA” study comes out of the blue here. This should have been mentioned in the introduction. The same holds with the “Other Rare Diseases Study PPI group” mentioned below.

See below

18. p6, line 27 – what are PPA collaborators? Patients, researchers, clinicians, caregivers, advocates? How are they selected?

To provide more context for the PPI work and the contribution of the Better Conversations with PPA PPI work to this study, as well as explaining the recruitment of PPI group members: the following has been added to this section:

“In line with guidance from the National Institute for Health and Care Research PPI work allows researchers to do research with people ‘with’ rather than ‘to’ people lived experience (<https://www.nihr.ac.uk/documents/ppi-patient-and-public-involvement-resources-for-applicants-to-nihr-research-programmes/23437>). The development of this study proposal was informed by PPI work undertaken during the Better Conversations with PPA (BCPPA) randomised controlled pilot study [9].

PPI collaborators identified that measurement tools needed to reflect what was important to people living with PPA [22]. The current COS-PPA study is overseen by the BCPPA and Other Rare Diseases Study PPI group. Members of this PPI group were invited from the Rare Dementia Support PPA Support Group (<https://www.raredementiasupport.org/>) using purposeful recruitment strategies.”

19. p6, line 38 – what is the “first author’s current fellowship award”?

The following has been added to provide more clarity that the award is a funded fellowship:
“first author’s current NIHR funded fellowship award.”

20. p6, line 43 – I am not sure what is “this protocol paper”.

p6, line 60 – Up until now I had the impression that the three studies mentioned in the abstract were already conducted and would be discussed here. However, only here I understood that this is “a protocol” and therefore that there are no results provided. I am aware that the title includes the word “protocol”. However, my expectation as a reader was much different.

We believe these concerns have been addressed by responses to points 10 and 11.

21. p7, line 12 – what are “collaborators” here? Are the same than those in page 6 line 27?

This subtitle has been amended for clarity:

“Collaborators for stages 2 & 3 of the study”

22. p7, line 17 – please change “Triallists” for “Trialists”. Which working group from the Collaboration of Aphasia Trialists did you contact? Did they discuss this in their regular meetings? I am aware that some groups are much more active than others. Also, the collaboration worked rather differently before and after it received funds from the European Union.

As a member of CATS the author contact the executive board, who disseminated it via relevant working groups. The following has therefore been added:

“These networks will be asked, via their executive committees and leads to disseminate information about the study to all relevant working groups.”

23. p7, line 50 – why is this section highlighted in grey?

Apologies but we are unsure what the reviewer is referring to as nothing in the document is highlighted in grey by the submitting authors.

24. p7, line 56 – how many people with will PPA enter the study? What will be their expected demographic details? Any chance that the diagnosis will be mixed? Would any people be included who had been recently diagnosed and had mild impairments or people who had severe impairments? We have not provided an anticipated recruitment number, but have outlined the number of participating countries. Additionally, the following details have been added to provide more specific information on demographic details to be collected:

“Collaborators will collate demographic data from participants on age, sex, type of PPA (including nfvPPA, svPPA, lvPPA and no specific variant of PPA diagnosed) and time since diagnosis. We anticipate this will include both mildly and more severally affected people with PPA.”

25. p9, line 29 – it is unclear why the authors mention that they want to replicate a study that is not published (under review). Can you truly replicate an unpublished work?

The paper is now published and the reference has been updated.

26. p9, line 41 – please specify what are “non-English measures”. Does this refer to the language in which the papers are written, or the language in which the assessments were conducted?

This has been amended as follows:

“Papers/measures not published in English will not be excluded from the study. “

27. p10, line 7 – It is unclear why the authors do not consider the possibility that studies do not

indicate PPA variant or mixed variants. Also, some studies may not discuss PPA variant à la Gorno-Tempini, but following previous classifications by Kertesz or Mesulam. Would the authors be willing to establish some kind of parallel between classifications? What to do in cases where apraxia of speech is or not considered compulsory?

Many thanks for this suggestion, the text has been updated to reflect this:

“Data will be extracted from each article with regard to which PPA variant measures were used with (lvPPA, svPPA, nfvPPA or no specific PPA variant, taking into account current [1] and previous PPA classification criteria [36,37]).”

28. p10, line 60 – “Participants will complete consent procedures required by each collaborator’s institution”. It is unclear how many institutions will end up participating and under which ethic approval(s) will this project be performed, if at all? Perhaps the main author has an umbrella approval that is potentially valid in other countries? It is unclear how the privacy of the participants will be protected, how the documents that will be used will be collected and/or destroyed. Perhaps all information gathered will be anonymous or pseudonymized? Later on, it is indicated that the data will be shared in anonymized format to the main author.

p11, line 42 – Information on ethics approval appears incomplete. Will this information be included in the eventual study?

The manuscript outlines all the ethical approvals across all institutions under the subtitles “Ethical Approvals” - that have already been secured, in line with journal requirements. Perhaps the reviewer is unable to view these as the manuscript has been anonymised for review?

VERSION 2 – REVIEW

REVIEWER	Mitchell, Claire Manchester University
REVIEW RETURNED	08-Mar-2024
GENERAL COMMENTS	Many thanks for your revisions these have addressed all the points I made. Best wishes for the work, I look forward to reading your findings.